# Prevalence and consequences of non-adherence to an evidence-based approach for incidental pulmonary nodules

**Max T. Wayne** [1]*, **Hallie C. Prescott** [1,2], **Douglas A. Arenberg** [1]

**1** Division of Pulmonary and Critical Care Medicine, Department of Internal Medicine, University of Michigan, Ann Arbor, MI, United States of America, **2** VA Center for Clinical Management Research, Ann Arbor, MI, United States of America

* maxtwayne@gmail.com

**Data Availability Statement:** All relevant data are within the paper and its Supporting Information files.

## Abstract

### Importance

Distinguishing benign from malignant pulmonary nodules is challenging. Evidence-based guidelines exist, but their impact on patient-centered outcomes is unknown.

### Objective

To understand if the evaluation of incidental pulmonary nodules that follows an evidence-based management strategy is associated with fewer invasive procedures for benign lesions and/or fewer delays in cancer diagnosis.

### Design

Retrospective cohort study.

### Setting

Large academic medical center.

### Participants

Adults ($\geq$18 years age) with an incidental pulmonary nodule discovered between January 2012 and December 2014. Patients with calcified nodules, prior nodules, prior diagnosis of cancer, high suspicion for pulmonary metastasis, or limited life expectancy were excluded.

### Exposure

Nodule management strategy (pre-specified based on evidence-based practices).

### Outcome

Composite of any invasive procedure for a benign nodule or delay in diagnosis in patients with cancer (>3 month delay once probability of cancer was >15%).

**Funding:** DAA - received grant from Michigan Institute for Clinical and Health Research (MICHR) grant UL1-TR000433 which supported this project. The funder had no role in study design, data collection and analysis, decision to publish, or preparation of the manuscript.

**Competing interests:** The authors have declared that no competing interests exist.

**Abbreviations:** ACCP, American College of Chest Physicians; BMI, body mass index; CI, confidence interval; CT, computed tomography; EHR, electronic health record; IQR, interquartile range; mm, millimeter; Pca, pre-test probability of cancer; PET, positron emission tomography; OR, odds ratio; SD, standard deviation.

## Results

Of 314 patients that met inclusion criteria, median age was 61, 46.5% were men, and 66.5% had current or former tobacco use. The mean nodule size was 10.3 mm, mean probability of cancer was 11.8%, and 14.3% of nodules were malignant. Evaluation followed an evidence-based strategy in 245 patients (78.0%), and deviated in 69 patients (22%). The composite outcome occurred in 26 (8.3%) patients. Among patients whose nodule evaluation was concordant with an evidence-based evaluation, 6.1% (15/245) experienced the composite outcome versus 15.9% (11/69) of patients with an evaluation that deviated from evidence-based recommendations ($P$<0.01).

## Conclusions and relevance

At a large academic medical center, more than 1 in 5 patients with an incidental pulmonary nodule underwent evaluation that deviated from evidence-based practice recommendations. Nodule evaluation that deviated from an evidence-based strategy was associated with biopsy of benign lesions and delays in cancer diagnosis, suggesting a need to improve guideline uptake.

## Introduction

Pulmonary nodules are common, with at least 1.5 million nodules discovered annually [1]. As the use of cross-sectional imaging increases, this number will continue to rise [2–6]. While more than 96% of lung nodules are benign, distinguishing benign from malignant nodules is challenging [7]. As a result, guidelines for the management of incidentally discovered pulmonary nodules have been developed to improve patient-centered outcomes [8–10]. Important outcomes in the management of pulmonary nodules include prompt diagnosis of malignancy while minimizing invasive procedures for patients with benign nodules. Strategies to maximize each of these outcomes are often in direct opposition, such that maximizing one outcome (*e.g.*, detection of cancer) may result in poor quality when measuring the other outcome (*e.g.*, invasive biopsies of benign lesions).

Prior studies have demonstrated that non-adherence to published guidelines in the management of pulmonary nodules is common (up to 40% depending on the setting) [11–13]. Patient preferences, individual risk factors, and system factors all may impact the management of incidentally discovered pulmonary nodules [13–17]. There is, however, a dearth of evidence that examines whether or how management of nodules that mirrors the approach outlined in evidence-based guidelines affects clinically relevant patient-centered outcomes. We reasoned that, if care that follows an evidence-based approach is not associated with fewer delays in cancer diagnosis or biopsies of benign processes, then the guidelines and the evidence behind them should be re-evaluated. On the other hand, if nodule management that mirrors evidence-based guidelines is associated with fewer delays in cancer diagnosis and less biopsies of benign processes, focusing on implementation of guidelines and the development of quality metrics based on these guidelines would be justified.

In this cohort study of patients with incidentally discovered pulmonary nodules, we classified each patient's evaluation as concordant or discordant with an evidence-based approach, then tested whether a guideline-concordant evaluation was associated with fewer delays in cancer diagnosis and biopsies of benign nodules. We hypothesized that a management strategy

that was concordant with evidence-based guidelines would result in fewer delays in cancer diagnosis and less invasive procedures for benign lesions.

## Materials and methods

### Study setting

This was a single-center, retrospective observational cohort study of patients with incidentally discovered pulmonary nodules identified and managed at the University of Michigan Medical Center. The study was approved with a waiver of informed consent by the University of Michigan institutional review board (HUM00111401). This study follows the Strengthening the Reporting of Observational Studies in Epidemiology (STROBE) reporting guideline for observational studies [18].

### Cohort identification and patient selection

We identified all patients with newly identified, incidentally discovered pulmonary nodules between January 2012 –December 2014. This time frame was chosen to coincide with the system wide implementation of a fully embedded electronic health record (EHR) (MiChart, Epic, Verona, WI [19,20]), to allow for extended follow-up time to completely assess outcomes of nodule evaluations, and because during this time-period there was no structured nodule management program (so as to avoid any bias in management).

Patients aged 18 years of age or older were identified by diagnostic codes from radiology reports and healthcare encounters [21]. The specific diagnostic codes used were *International Classification of Diseases*, *Ninth Revision*, *Clinical Modification* codes 793.11 (solitary pulmonary nodule) and 793.19 ("other non-specific abnormal finding of lung field") [22,23]. Patients with calcified nodules, a diagnosis of cancer within the prior 5 years (excluding non-melanoma skin cancer or prostate cancer), previously identified nodules, had less than 12 months life expectancy (as indicated by chart notes documenting recognition of, but no intention to evaluate, the incidental nodule), a high suspicion for lung metastasis (as indicated in the radiology report), or those in whom long-term follow-up could not be measured (*e.g.*, patient elected to have the nodule managed by a physician outside of the health system) were excluded.

### Data collection

After identifying all patients with an eligible diagnosis code, patients were randomly sorted and a random sample of 800 patients was selected for data extraction. Collected data included all elements of the Brock nodule calculator, patient demographics and clinical characteristics, physician diagnosis of chronic obstructive pulmonary disease, nodule characteristics (size, attenuation, spiculation, solid vs ground glass, upper versus lower/middle lobe), follow up radiologic imaging (computed tomography (CT) or positron emission tomography (PET)), referral to pulmonary and thoracic surgery, invasive procedures to investigate the nodule, and pathology results. Referral to pulmonary and thoracic surgery were identified if documented in the EHR.

### Primary exposure and outcome definition

The exposure of interest was the nodule evaluation strategy and whether it was concordant with an evidence-based approach. We defined an evidence-based approach according to the algorithm depicted in **Fig 1**. This strategy was prospectively defined using existing best practice recommendations [10]. The appropriate diagnostic path was therefore dependent upon first determining the probability of cancer (Pca). While in clinical practice, these management decisions are based on clinical intuition rather than formally calculating a probability of

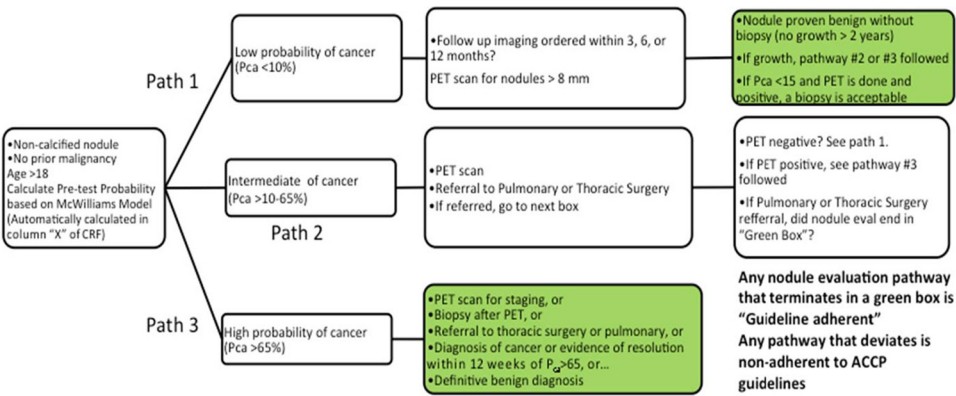

**Fig 1. Nodule evaluation pathway.** Flow diagram showing nodule evaluation pathway (modeled using the ACCP guidelines). The first step was to calculate a pre-test probability of cancer using the Brock (or McWilliams) model using clinical and nodule characteristics. Based on this pre-test probability, nodules were classified as low, intermediate, or high probability of cancer. Concordance with evidence-based guidelines resulted in a nodule evaluation pathway that resulted in a green box, otherwise the evaluation was determined to be non-adherent to guidelines. ACCP: American College of Chest Physicians; Pca: Pre-test probability of cancer; PET: Positron emission tomography.

malignancy, to retrospectively assess nodules in an unbiased manner, we used a previously validated prediction model to quantify risk of malignancy [24]. Accordingly, any deviation from the algorithm depicted in **Fig 1** was deemed to be an evaluation discordant with evidence-based recommendations. For the purpose of this manuscript, the design of this pathway was intentionally strict so that exposure and outcome could be retrospectively identified without bias.

Our composite primary outcome was either a delay in diagnosis of lung cancer or biopsy for benign disease. Delay in diagnosis of lung cancer was defined as >3 month delay in diagnosis after a nodule was detected with a probability of cancer of greater than 15%, and further classified as moderate (>3 to <6 month) versus severe delay (>6 months). The rationale for our definition of "delay" is based in the recommended pathway for guideline evaluation, as well as studies examining outcomes of nodule management. We chose a cut off of 15%, based on a study by Tanner, *et al.* [11] showing that none of the subjects with nodules having a Pca <15% were diagnosed with cancer over 2 years of follow up. We reasoned that a probability greater than 15%, according to current guidelines, should lead to additional investigation (*e.g.*, PET scan, or additional imaging). A negative PET scan would reduce the post-test Pca sufficiently to prompt a recommendation for radiographic follow-up, while a positive PET scan in that setting was considered an indication for referral for either biopsy, or surgical resection (see Fig 1 for the algorithm). The second component of our composite primary outcome was surgery or other invasive procedures (*i.e.*, transbronchial biopsy or transthoracic biopsy) for benign lung nodules. For nonsurgical biopsies, "benign" was defined both by the results of the biopsy as well as stability, or resolution over follow-up of at least 12 months' duration.

## Sample size and statistical analysis

To calculate adequate sample size, we assumed that the proportion of individuals having the composite outcome (invasive procedure for a benign lung nodule or diagnostic delay) would be 0.20 based on a study of pulmonologists' nodule evaluation [11]. In patients whose lung nodule evaluation did not follow our pre-defined algorithm, we predicted that the proportion of patients having the composite primary outcome (invasive procedure for a benign lung nodule or a diagnostic delay) would double to 0.4. Finally, we assumed a 3:1 ratio of patients

whose nodule evaluation was concordant with an evidence-based approach [12,25]. Using these assumptions, we expected to have 90% power to detect a 0.12 absolute increase in our primary outcome (0.2 to 0.32) if 200 patients were included. Given that patients were identified using *ICD-CM* codes which would not account for other inclusion/exclusion criteria, we planned a preliminary analysis after review of 40 charts to assess how many would need to be reviewed to meet our target. Based off this preliminary analysis, which showed that approximately 75% of reviewed charts met exclusion criteria, to reach our target of 200 patients, we selected 800 patients for full review.

We tested for differences using a chi-square test for categorial variables and Mann Whitney U test for continuous variables. Secondary outcomes included the length of delay (moderate or severe) and type of invasive procedure (surgical versus non-surgical). We also noted (for quality improvement) individuals who were lost to follow-up entirely. In exploratory analysis, univariable logistic regression was performed to identify factors associated with the composite outcome of interest. Additionally, multivariable logistic regression was performed to assess the relationship between the composite outcome and the probability of cancer and nodule management strategy. Statistical analyses were performed using STATA/MP version 17.0 (StataCorp, College Station, TX). We considered $p < 0.05$ (two-sided) to be significant.

## Results

We identified 9,404 patients with a nodule diagnosis code based on our search strategy, of whom 800 (8.5%) were selected at random for review. Of these, 314 (39.3%) met study criteria and were included in our analysis (Fig 2). The most common reasons for exclusion were prior

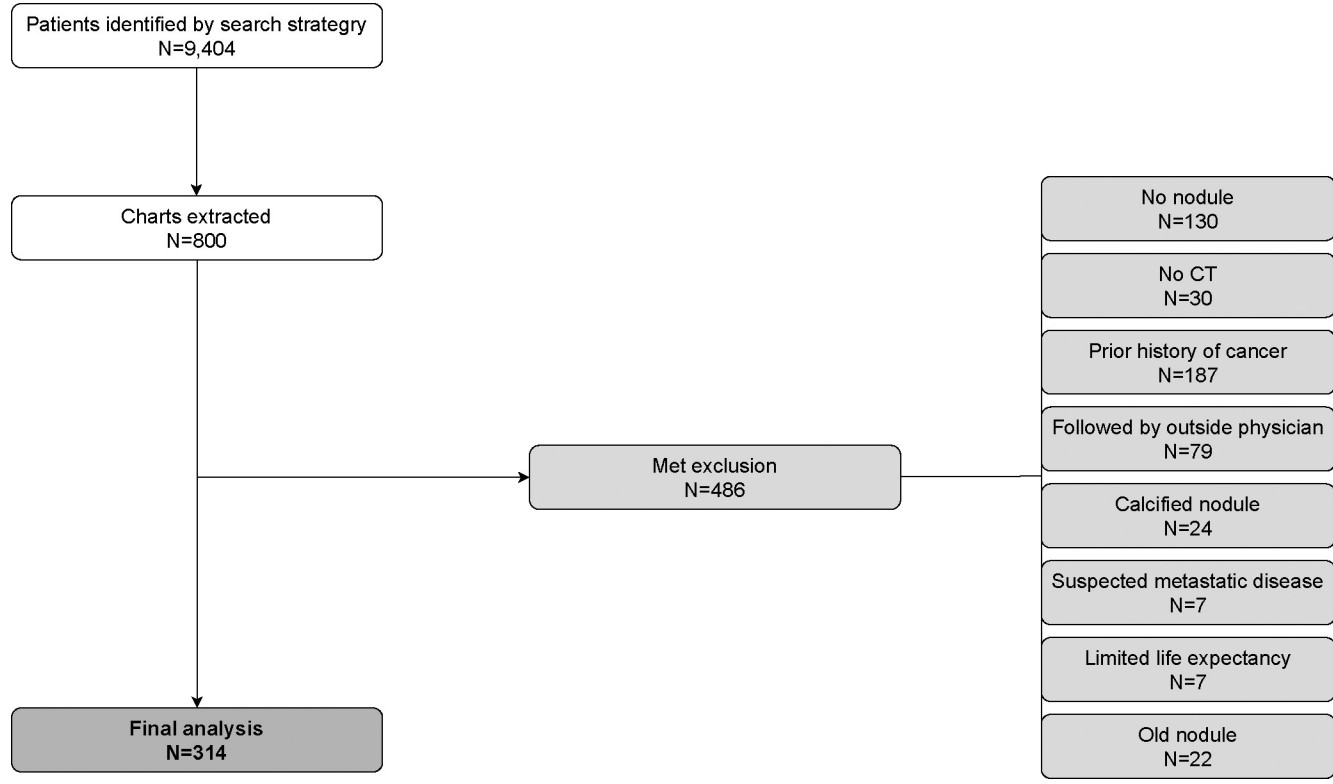

**Fig 2. Flow diagram.** Flow diagram showing identification of our cohort included for final analysis. Our search strategy identified 9,404 unique patients, 800 were extracted for chart review, and a total of 314 met criteria for final analysis.

**Table 1. Characteristics of patients and nodules by adherence to ACCP guidelines.**

| | | Overall N = 314 | Adherent N = 245 | Non-adherent N = 69 | *P-value* |
|---|---|---|---|---|---|
| Median Age (IQR) | | 61 (51–70) | 60 (51–69) | 62 (54–70) | 0.22 |
| Gender, N (%) | | | | | |
| | Male | 146 (46.5%) | 110 (44.9%) | 36 (52.2%) | 0.28 |
| | Female | 168 (53.5%) | 135 (55.1%) | 33 (47.8%) | |
| BMI, median (IQR) | | 28.4 (24.4–33.6) | 28.4 (24.4–33.2) | 28.0 (24.5–34.4) | 0.53 |
| Current or prior tobacco use, N (%) | | 208 (66.5%) | 161 (66.0%) | 47 (68.1%) | 0.74 |
| Median pack-years (among current or former users), (IQR) | | 10.0 (0.0–39.0) | 15.0 (0.0–40.0) | 5.0 (0.0–27.5) | 0.21 |
| Family history of cancer, N (%) | | 48 (15.3%) | 44 (18.0%) | 4 (5.8%) | 0.01 |
| COPD or emphysema, N (%) | | 115 (36.6%) | 96 (39.2%) | 19 (27.5%) | 0.08 |
| Nodule size in mm, mean (SD) | | 10 (11) | 11 (12) | 8 (6) | 0.09 |
| Nodule size in mm, median (IQR) | | 6 (4–12) | 6 (4–13) | 6 (5–8) | 0.58 |
| Upper lobe nodule, N (%) | | 166 (52.9%) | 130 (53.1%) | 36 (52.2%) | 0.90 |
| Spiculated nodule, N (%) | | 43 (13.7%) | 37 (15.1%) | 6 (8.7%) | 0.17 |
| Nodule type, N (%) | | | | | |
| | Solid | 251 (80.2%) | 198 (81.1%) | 53 (76.8%) | 0.58 |
| | Part solid | 51 (16.3%) | 37 (15.2%) | 14 (20.3%) | |
| | Ground glass | 11 (3.5%) | 9 (3.7%) | 2 (2.9%) | |
| Referral, N (%) | | | | | |
| | Pulmonary | 170 (54.1%) | 138 (56.3%) | 32 (46.4%) | 0.14 |
| | Thoracic surgery | 42 (13.4%) | 34 (13.9%) | 8 (11.6%) | 0.62 |
| Procedures, N (%) | | | | | |
| | PET scan | 87 (27.7%) | 70 (28.6%) | 17 (24.6%) | 0.52 |
| | Biopsy (non-surgical* or surgical) | 63 (20.1%) | 52 (21.2%) | 11 (15.9%) | 0.33 |
| | Surgery | 28 (8.9%) | 21 (8.6%) | 7 (10.1%) | 0.69 |
| Probability of cancer, mean (SD) | | 11.8% (21.0) | 13.2% (22.2) | 7.0% (14.9) | 0.03 |
| Nodule confirmed malignant, N (%) | | 44 (14.0%) | 39 (15.9%) | 5 (7.2%) | 0.07 |

*Non-surgical biopsy included bronchoscopic and CT-guided biopsies.

BMI: Body mass index; CT: Computed tomography; IQR: Interquartile range; mm: Millimeter; SD: Standard deviation.

history of cancer (23.4%, n = 187), no nodule on CT scan (16.3%, n = 130), and nodule followed by physician within a different health system (9.9%, n = 79).

Of the 314 included patients, 270 (86.0%) had a nodule that was ultimately deemed benign, and 44 (14.0%) had a nodule that was determined to be malignant (**Table 1**). 245 (78.0%) had an evaluation that was concordant with our pre-defined algorithm modeled on evidence-based guidelines, as defined by Fig 1, and 69 (22.0%) were evaluated in a manner that deviated from the path in Fig 1 and were categorized as discordant. The most common reason for discordance to an evidence-based evaluation was a lack of appropriate follow-up CT imaging (n = 58; 84.1% of non-adherent evaluations).

Median age was 61 years, 146 (46.5%) were male, and 168 (53.5%) were female (**Table 1**). Age, sex, smoking status, and nodule size did not differ between patients with concordant versus discordant evaluations (**Table 1**). Among all patients, 170 (54.1%) were seen by a pulmonologist and 42 (13.4%) were seen by a thoracic surgeon.

The mean pre-test probability of cancer was 11.8% (**Table 1**). Among nodules with a pre-test probability of cancer <5%, 2 (1.0%) were malignant, and among nodules with a pre-test

probability of cancer <10%, 3 (1.3%) were malignant. 87 (27.7%) patients underwent PET scan, of whom 18 had low (<10%) pre-test probability of malignancy, 53 had intermediate (10–65%) pre-test probability of malignancy, and 16 had high (>65%) pre-test probability of malignancy. Among patients with negative PET scans (n = 17; as defined by the interpreting radiologist), none were malignant. Of the 51 patients with positive PET scans, 39 (73.6%) of the nodules were malignant. Of the 17 patients with intermediate PET scans, 3 (17.6%) of the nodules were malignant.

Among all patients, 26 (8.3%) met the composite primary outcome, either a delay in cancer diagnosis of at least 3 months after probability of malignancy was >15% (n = 5, 1.6%) or an invasive procedure for a benign process (n = 21, 6.7%). The most common diagnoses for patients who underwent an invasive procedure for benign processes were sarcoidosis (n = 9, 43%) and infection (n = 4, 19%) (S1 Table). The composite outcome occurred in 6.1% (15/245) of patients with an evaluation that followed our pre-defined algorithm vs 15.9% (11/69) of patients whose evaluation was discordant with the pre-defined evidence-based approach, $P<0.01$ (Fig 3). No patients with a nodule evaluation that was concordant with an evidence-based strategy experienced a delay in cancer diagnosis of at least 3 months when there was moderate risk of cancer, whereas 5 patients (7.2%) with an evaluation that was discordant from the evidence based algorithm experienced a delay in diagnosis ($P<0.001$) (Fig 3 & Table 2).

In univariable logistic regression analysis, an evidence-based evaluation was associated with lower odds of the composite outcome (OR = 0.34; 95% CI: 0.15–0.79). Larger nodule size, spiculation, intermediate nodule pre-test probability of cancer, referral to pulmonary, referral to thoracic surgery, obtaining a PET scan, and PET scan with either intermediate or positive test results were each associated with increased odds of the composite outcome (**Table** 2). In multivariable logistic regression, probability of malignancy was associated with increased odds of the composite outcome (OR = 1.03; 95% CI: 1.02–1.05) whereas an evidence-based approach was associated with decreased odds of the composite outcome (OR = 0.14; 95% CI: 0.07–0.28).

## Discussion

In this cohort study of over 300 patients with an incidentally discovered pulmonary nodule, more than 1 in 5 patients underwent a nodule evaluation that was discordant with an evidence-based approach to nodule evaluation. Moreover, nearly 1 in 10 patients experienced either at least a 3-month delay in cancer diagnosis once there was moderate probability of malignancy or underwent a biopsy of a benign process. Nodule evaluations that followed an evidence-based algorithm were associated with lower odds of this composite outcome.

Our results differ somewhat from previous studies of lung nodule management. Tanner et al [11] studied patients who had seen a pulmonologist, and the study population included patients seen at 18 geographically diverse community practices. Important differences include that our study focused on patients identified because of the finding of a nodule on a radiology report and therefore included patients not referred specifically for nodule evaluations. Likely because of this, the prevalence of malignancy in this study (14% of 314) was lower than reported by Tanner and colleagues [11] (25% of 377). Secondly, our cohort completed their nodule evaluation at a tertiary care teaching hospital. Though Tanner et al did not specifically identify guideline-concordant versus discordant evaluations, comparisons can be inferred for specific outcomes. For example, in the Tanner study, 44% of low-risk patients underwent one or more invasive procedures for a benign nodule. This outcome was less frequent in our study, but was more common when management pathways diverged from an evidence-based pathway.

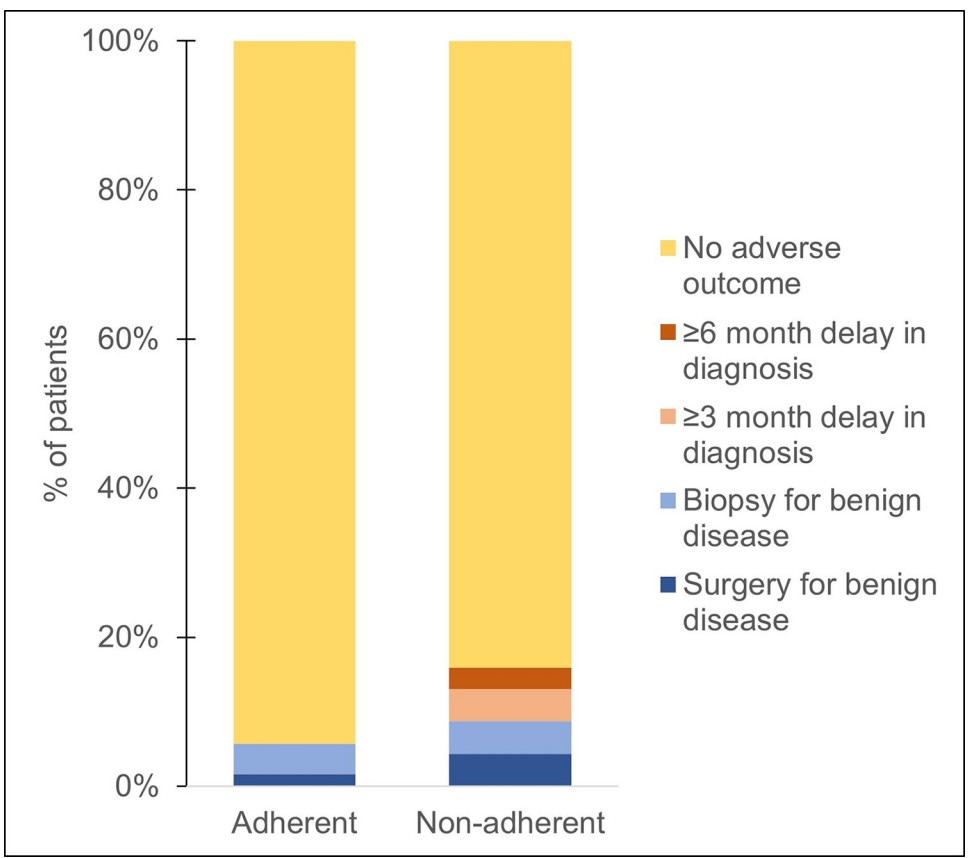

**Fig 3. Outcomes of nodule evaluation by adherence to ACCP guidelines.** Bar graph showing outcomes of nodule evaluation according to whether nodule management strategy followed an evidence-based approach. Of the 314 patients included, 245 (78.0%) were concordant with an evidence-based approach and 69 (22.0%) were discordant. Among patients with an evidence-based evaluation, 15 patients (6.1%) met the primary composite outcome: 4.1% underwent a non-surgical biopsy for benign disease and 2.0% underwent surgery for benign disease. Among patients with an evaluation discordant with evidence-based recommendations, 11 (15.9%) met the primary composite outcome: 4.3% non-surgical biopsy for benign disease; 4.3% surgery for benign disease; 4.3% >3 month delay in diagnosis once the probability of malignancy was >15%; and 2.9 % ≥ 6 month delay in diagnosis once the probability of malignancy was >15%. The difference between concordant and discordant evaluation groups was statistically significant ($P<0.01$).

**Table 2. Outcomes of nodule evaluation based on concordance versus discordance with guideline recommended practice.**

| | Guideline concordant evaluation N = 245 | Guideline discordant evaluation N = 69 | *P-value** |
|---|---|---|---|
| Composite primary outcome, N (%) | 15 (6.1%) | 11 (15.9%) | 0.01 |
| Invasive procedure for benign disease | | | |
| Surgical or non-surgical biopsy for benign disease | 15 (6.1%) | 6 (8.7%) | 0.45 |
| Surgery for benign disease | 4 (1.6%) | 3 (4.3%) | 0.18 |
| Non-surgical biopsy for benign disease** | 11 (4.5%) | 4 (5.8%) | 0.66 |
| Delay in diagnosis | | | |
| Any delay in diagnosis*** | 0 (0%) | 5 (7.2%) | <0.001 |
| Moderate delay (>3 months & <6 months) | 0 (0%) | 3 (4.3%) | 0.001 |
| Severe delay (>6 months) | 0 (%) | 2 (2.9%) | <0.01 |

*P-value* calculated as chi-squared difference between adherent and non-adherent groups.

**Non-surgical biopsy included either transthoracic needle biopsy or bronchoscopic biopsy.

***Any delay in diagnosis defined as >3 months from time probability of malignancy was >15% to diagnosis (calculated using Brock model).

Similarly, Wiener et al [26] evaluated adherence to guidelines in the US Veterans Affairs system, finding that, among patients with a screen-detected nodule, 44.7% received care inconsistent with LUNG-Rads follow-up recommendations. They reported that 17.8% of patients experienced over-evaluation, defined as testing more frequently or for longer duration than recommended, or performance of tests outside of recommendations (*e.g.*, PET or biopsy for nodules <8 mm). Additionally in this study, 26.9% experienced evaluations that were less stringent than recommended ("undervaluation") characterized by delays in, or failure to perform, radiographic surveillance, which was similar to our finding that 18% of patients did not receive recommended radiographic surveillance [26].

In a separate study, the same investigators prospectively observed patient adherence to clinician recommendations (defined as receiving the follow-up scan within 30 days of the recommended date) and clinician adherence to guidelines (defined as requesting the follow-up scan within 30 days of the recommended date) [12]. They identified important factors associated with greater adherence (e.g., *High-quality communication* as defined by a validated Consultation Care Measure scale) or lesser adherence (e.g., *Distress* as measured by the Impact of Event Scale).

While prior studies examined how frequently nodule guidelines are followed [11,13,26], there is limited data on whether invasive procedures for benign lesions or diagnostic delays are minimized by guideline adherence (or, conversely, whether these adverse outcomes are more common in the setting of non-adherent evaluation). This study provides indirect support for the existing ACCP guidelines in that patients whose evaluations deviated from the pathway (based upon ACCP guidelines) were more likely to experience a composite outcome that included an invasive procedure for benign lesions or delays in cancer diagnosis. It further confirms and highlights that a significant proportion of patients fail to receive guideline-recommended evaluation, even within a tertiary care academic medical center with a multidisciplinary team and dedicated nodule clinic.

Patients referred to either a pulmonologist or thoracic surgeon were more likely to receive an invasive procedure. However, and somewhat surprisingly, referral to either a pulmonologist or thoracic surgeon was not associated with greater adherence to an evidence-based approach to nodules and referral to these providers was associated with our composite outcome. One possible explanation for this association is that pulmonologists and thoracic surgeons are frequently the providers obtaining tissue diagnosis, so if patients are referred to specialists late, we would identify this association even though these providers were not directly responsible for the delay in diagnosis. Similarly, if tissue was obtained to confirm a diagnosis of sarcoidosis, this would be classified as meeting the composite outcome according to our definition, even though it may have provided clinically helpful information. Consequently, this association may have been influenced by our study design. Nevertheless, our findings suggest the need to focus on implementation strategies that promote guideline adherence, even among specialists evaluating pulmonary nodules.

Our study has several limitations. There is no gold-standard to identify pulmonary nodules in the electronic health record. We used ICD-9 codes to identify nodules, which may not have captured all patients. It is also unclear if this identifies an unbiased sample. Due to the retrospective design, we were unable to evaluate the appropriateness of the decision to pursue an invasive procedure, so it is possible that delays in diagnosis or biopsies of benign nodules could have been appropriate based on the clinical context. Third, it is possible we excluded confounders that may have explained a referral pattern to pulmonary or thoracic surgery. Fourth, given our study design, we were unable to assess the relationship between specific factors (*e.g.*, patient anxiety, family history, communication strategies) and nodule management strategies and decisions to pursue an invasive procedure, nor were we able to understand if

non-adherence was driven by patient-preference or physician-related practice. Finally, this study was performed at a single academic medical center, which may limit the generalizability of the findings.

However, this study also has a number of strengths. By using a fully embedded electronic health record, we were able to track all testing done for the nodule in question. This cannot be done easily with administrative databases, although they make it easier to track costs and charges, which we have not attempted. Importantly, we used prevailing guidelines to establish a clear framework for nodule evaluation that matches best practice recommendations. And critically, this study provides evidence to support current nodule guidelines and highlights the need for more widespread adoption and uptake.

## Conclusions

At a single academic medical center, nearly 1 in 5 incidentally discovered pulmonary nodules did not follow an evaluation that was concordant with an evidence-based approach. Nodule evaluation that followed evidence-based recommendations was associated with reduced risk of delayed diagnosis of lung cancer or biopsy of benign diseases. Future research should focus on dissemination and development of quality metrics based on these guidelines.

## Supporting information

**S1 Table. Invasive procedures for benign processes.**
(DOCX)

**S2 Table. Factors associated with an adverse outcome.**
(DOCX)

**S1 File. This is the dataset used for analysis.**
(CSV)

## Author Contributions

**Conceptualization:** Max T. Wayne, Douglas A. Arenberg.

**Data curation:** Max T. Wayne, Douglas A. Arenberg.

**Formal analysis:** Max T. Wayne, Hallie C. Prescott, Douglas A. Arenberg.

**Funding acquisition:** Douglas A. Arenberg.

**Investigation:** Max T. Wayne, Hallie C. Prescott, Douglas A. Arenberg.

**Methodology:** Max T. Wayne, Hallie C. Prescott, Douglas A. Arenberg.

**Project administration:** Max T. Wayne, Douglas A. Arenberg.

**Resources:** Douglas A. Arenberg.

**Supervision:** Hallie C. Prescott, Douglas A. Arenberg.

**Validation:** Hallie C. Prescott, Douglas A. Arenberg.

**Writing – original draft:** Max T. Wayne.

**Writing – review & editing:** Max T. Wayne, Hallie C. Prescott, Douglas A. Arenberg.

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
