## [Decision Letter · Decision Letter 0]

30 Jun 2022

PONE-D-22-04795Prevalence and consequences of non-adherence to an evidence-based approach for incidental pulmonary nodulesPLOS ONE

Dear Dr. Wayne,

Thank you for submitting your manuscript to PLOS ONE. After careful consideration, we feel that it has merit but does not fully meet PLOS ONE’s publication criteria as it currently stands. Therefore, we invite you to submit a revised version of the manuscript that addresses the points raised during the review process.

Please note that we have only been able to secure a single reviewer to assess your manuscript; their comments are appended below, and should be addressed in full. We are issuing a decision on your manuscript at this point to prevent further delays in the evaluation of your manuscript. Please be aware that the editor who handles your revised manuscript might find it necessary to invite additional reviewers to assess this work once the revised manuscript is submitted. However, we will aim to proceed on the basis of this single review if possible.

We look forward to receiving your revised manuscript.

Kind regards,

Emily Chenette

Editor in Chief

PLOS ONE

Journal Requirements:

Reviewers' comments:

Reviewer's Responses to Questions

**Comments to the Author**

1. Is the manuscript technically sound, and do the data support the conclusions?

Reviewer #1: Yes

2. Has the statistical analysis been performed appropriately and rigorously? 

Reviewer #1: Yes

3. Have the authors made all data underlying the findings in their manuscript fully available?

Reviewer #1: Yes

4. Is the manuscript presented in an intelligible fashion and written in standard English?

Reviewer #1: Yes

5. Review Comments to the Author

Reviewer #1: The study analyzes the outcomes of non-adherence to guidelines on management of pulmonary nodules. It is single institution analysis and is performed on 314 patients. The paper is well written.

It is not unexpected to achieve such results with non-adherence to management guidelines. The study does not provide sub analysis of the data. Development of a composite score is important and the authors should be commended for their approach.

My comments are below.

1. Although the variables are included in the composite score, what was the difference in adherence to guidelines within age groups. <55 and ≥55? As we all know, 55 is an important age limit for screening.

2. Similarly, were there any difference in adherence to guidelines secondary to size of the nodule? <10 mm versus ≥10 mm.

3.

4. Family history of cancer was significantly high in the adherent group. Would this have been a motivating factor for adherence to the follow-up protocol?

5. Non adherent group patients had less COPD, less amount of smoking (pack years) and relatively smaller nodules. Spiculation was less and probability of cancer was lower. Would these have been a factor for non-adherence? Do the authors think that the communication method with the patient would affect non-adherence?

6. Patient anxiety is sometimes an important factor. What was the percentage were due to patient anxiety in benign nodules that underwent intervention?

7. Is there anyway to identify if non-adherences were Physician or patient related?

Overall the discussion is thorough and tables figures are adequate.

6. PLOS authors have the option to publish the peer review history of their article (what does this mean?). If published, this will include your full peer review and any attached files.

Reviewer #1: **Yes: **Hasan Batirel

---

## [Author Response · Author response to Decision Letter 0]

6 Jul 2022

July 2, 2022

Emily Chenette, PhD

Editor in Chief, PLOS ONE

Dear Dr. Chenette:

Please accept our revised manuscript “Prevalence and consequences of non-adherence to an evidence-based approach for incidental pulmonary nodules” (PONE-D-22-04795) by Max T. Wayne, Hallie C. Prescott, & Douglas A. Arenberg for publication as an original research article in PLOS ONE.

We appreciate the editor and reviewer comments, which have helped us to further strengthen the manuscript. 

Reviewer 1

C1: Although the variables are included in the composite score, what was the difference in adherence to guidelines within age groups. <55 and ≥55? As we all know, 55 is an important age limit for screening.

R1: There was no difference in adherence to guidelines when we stratified patients by age groups (less than 55 versus 55 or older). Among patients less than 55 years of age, 82.7% were adherent versus 75.7% among patients 55 years or older (p=0.16).

However, for this project we were specifically looking at incidental, not screen detected, nodules and additionally, patients were included from an era that preceded screening so the age cut-off would also not apply. 

Age did impact cancer risk in these incidental nodules, and this was incorporated in the calculated probability of cancer.

To better acknowledge this fact, our results now reads: “Age, sex, smoking status, and nodule size did not differ between patients with concordant versus discordant evaluations.”

C2: Similarly, were there any difference in adherence to guidelines secondary to size of the nodule? <10 mm versus ≥10 mm.

R2: Similarly, there was no difference in adherence to guideline concordant nodule strategy based on nodule size. Among nodules less than 10 mm in size, 75.7% were adherent versus 83.3% among nodules 10 mm or greater (p=0.13)

Our results now reads: “Age, sex, smoking status, and nodule size did not differ between patients with concordant versus discordant evaluations.”

C3: Family history of cancer was significantly high in the adherent group. Would this have been a motivating factor for adherence to the follow-up protocol?

R3: This is an interested point. Given our study design, we cannot definitively say whether family history was a motivating factor for adherence but it certainly would make sense that in individuals with a family history of lung cancer, there would be added motivation to follow these nodules.

We have amended our limitations to reflect this. It now reads: “Fourth, given our study design, we were unable to assess the relationship between specific factors (e.g., patient anxiety, family history, communication strategies) and nodule management strategies and decisions to pursue an invasive procedure. . .”

C4: Non adherent group patients had less COPD, less amount of smoking (pack years) and relatively smaller nodules. Spiculation was less and probability of cancer was lower. Would these have been a factor for non-adherence? 

R4: Unfortunately, based on our study design, it is hard to know for certain if these factors were the drivers in non-adherence but again it makes sense that the motivation to follow a guideline-concordant strategy would be less for patients with lower risk nodules and therefore these factors may have played a role in driving this non-adherence.

Given this limitation, we have amended our discussion to reflect that given our study design, we were unable to assess how specific factors impacted management strategies. Our limitations now reads: “Fourth, given our study design, we were unable to assess the relationship between specific factors (e.g., patient anxiety, family history, communication strategies) and nodule management strategies and decisions to pursue an invasive procedure. . .”

C5: Do the authors think that the communication method with the patient would affect non-adherence?

R5: This is an interesting question and should be considered in prospective studies aiming to improve adherence as this could easily have impacted the patient’s (or provider’s) willingness to pursue additional testing. Based on our study design, we can only speculate that this may have impacted adherence and non-adherence.

Given this, we have amended our discussion to reflect this limitation. The text now reads: “Fourth, given our study design, we were unable to assess the relationship between specific factors (e.g., patient anxiety, family history, communication strategies) and nodule management strategies and decisions to pursue an invasive procedure. . .”

C6: Patient anxiety is sometimes an important factor. What was the percentage were due to patient anxiety in benign nodules that underwent intervention?

R6: We agree with the reviewer that this is an important question. Unfortunately, given our retrospective study design, we are unable to answer this question. However, this is an important question that future prospective studies should aim to answer. 

We have revised our limitations to acknowledge this fact. It now reads: “Fourth, given our study design, we were unable to assess the relationship between specific factors (e.g., patient anxiety, family history, communication strategies) and nodule management strategies and decisions to pursue an invasive procedure. . .”

C7: Is there anyway to identify if non-adherences were Physician or patient related?

R7: We agree that understanding if non-adherence is physician or patient-related is an important question to answer as developing strategies to fix this problem depend on the reason for non-adherence. However, given our retrospective study design and the limitations of the electronic health record, we are unfortunately unable to answer this question. 

We have revised our limitations to acknowledge this fact. It now reads: “Fourth, given our study design, we were unable to assess the relationship between specific factors (e.g., patient anxiety, family history, communication strategies) and nodule management strategies and decisions to pursue an invasive procedure, nor were we able to understand if non-adherence was driven by patient-preference or physician-related practice.”

Journal Comments

C1: Please ensure that your manuscript meets PLOS ONE's style requirements, including those for file naming. 

R1: We have reviewed the formatting and style requirements and adjusted our manuscript accordingly.

C2: In your Data Availability statement, you have not specified where the minimal data set underlying the results described in your manuscript can be found. PLOS defines a study's minimal data set as the underlying data used to reach the conclusions drawn in the manuscript and any additional data required to replicate the reported study findings in their entirety. All PLOS journals require that the minimal data set be made fully available. Upon re-submitting your revised manuscript, please upload your study’s minimal underlying data set as either Supporting Information files or to a stable, public repository and include the relevant URLs, DOIs, or accession numbers within your revised cover letter. For a list of acceptable repositories, please see http://journals.plos.org/plosone/s/data-availability#loc-recommended-repositories. Any potentially identifying patient information must be fully anonymized.

R2: We have uploaded our study’s minimal underlying data set as a Supporting Information file. This file contains a de-identified dataset that can be used to replicate the study findings.

C3: Your ethics statement should only appear in the Methods section of your manuscript. If your ethics statement is written in any section besides the Methods, please move it to the Methods section and delete it from any other section. Please ensure that your ethics statement is included in your manuscript, as the ethics statement entered into the online submission form will not be published alongside your manuscript. 

R3: Our ethics statement only appears in the Methods section of the manuscript as requested.

C4: Please review your reference list to ensure that it is complete and correct. If you have cited papers that have been retracted, please include the rationale for doing so in the manuscript text, or remove these references and replace them with relevant current references. Any changes to the reference list should be mentioned in the rebuttal letter that accompanies your revised manuscript. If you need to cite a retracted article, indicate the article’s retracted status in the References list and also include a citation and full reference for the retraction notice.

R4: We have reviewed our reference list to ensure completeness and correctness.

We appreciate the time and thoughtful comments of the reviewer and editor, and look forward to hearing back from you. We are willing to consider further modifications of the manuscript as necessary to optimize it for PLOS ONE’s discerning readership.

Sincerely,

Max Wayne, MD

Fellow, Department of Internal Medicine

Division of Pulmonary and Critical Care Medicine

University of Michigan

---

## [Editor Report · Decision Letter 1]

23 Aug 2022

Prevalence and consequences of non-adherence to an evidence-based approach for incidental pulmonary nodules

PONE-D-22-04795R1

Dear Dr. Wayne,

We’re pleased to inform you that your manuscript has been judged scientifically suitable for publication and will be formally accepted for publication once it meets all outstanding technical requirements.

Kind regards,

Ming-Ching Lee

Academic Editor

PLOS ONE
---

## [Editor Report · Acceptance letter]

1 Sep 2022

PONE-D-22-04795R1 

Prevalence and consequences of non-adherence to an evidence-based approach for incidental pulmonary nodules 

Dear Dr. Wayne:

I'm pleased to inform you that your manuscript has been deemed suitable for publication in PLOS ONE. Congratulations! Your manuscript is now with our production department. 

Kind regards, 

on behalf of

Dr. Ming-Ching Lee 

Academic Editor

PLOS ONE